# Lindblad Dynamics and Disentanglement in Multi-Mode Bosonic Systems

**DOI:** 10.3390/e23111409

**Published:** 2021-10-27

**Authors:** Alexei D. Kiselev, Ranim Ali, Andrei V. Rybin

**Affiliations:** 1Laboratory of Quantum Processes and Measurements, ITMO University, Kadetskaya Line 3b, 199034 Saint Petersburg, Russia; 2Faculty of Photonics, ITMO University, Kronverksky Pr. 49, bldg. A, 197101 Saint Petersburg, Russia; ranimali19932014@hotmail.com; 3Center of Information Optical Technology, ITMO University, Birzhevaya Line 14a, 199034 Saint Petersburg, Russia; andrei.rybin@itmo.ru

**Keywords:** open quantum systems, Lindblad equation, disentanglement

## Abstract

In this paper, we consider the thermal bath Lindblad master equation to describe the quantum nonunitary dynamics of quantum states in a multi-mode bosonic system. For the two-mode bosonic system interacting with an environment, we analyse how both the coupling between the modes and the coupling with the environment characterised by the frequency and the relaxation rate vectors affect dynamics of the entanglement. We discuss how the revivals of entanglement can be induced by the dynamic coupling between the different modes. For the system, initially prepared in a two-mode squeezed state, we find the logarithmic negativity as defined by the magnitude and orientation of the frequency and the relaxation rate vectors. We show that, in the regime of finite-time disentanglement, reorientation of the relaxation rate vector may significantly increase the time of disentanglement.

## 1. Introduction

The theory of open quantum systems is very important for describing the transfer and storage of quantum information. In quantum information theory, these processes are generally described in terms of completely positive trace-preserving maps known as the quantum channels (see, e.g., [1,2,3,4,5] for analysis of the mathematical structures related to the quantum channels). There is also a variety of master equations for the reduced density matrix derived using different assumptions and approximations [6,7,8,9,10,11].

In particular, within the Markov approximation, master equations can often be cast into the well-known Lindblad form [12,13,14] which preserves complete positivity of the dynamics. This equation is also sometimes referred to as the Gorini–Kossakowski–Sudarshan–Lindblad (GKSL) equation.

Though the general physics behind Lindblad-type master equations has been extensively discussed (see, e.g., References [15,16,17,18]), the mathematical techniques developed for the analysis of bosonic systems are mostly applicable to the single-mode Lindblad equation [19,20,21,22,23,24,25] and cannot be employed to treat its multi-mode generalizations (the most general form of the multi-mode Lindblad equation is described, e.g., in [26]).

In a recent paper [27], Fock-like eigenstates of a Lindbladian are constructed using Lie algebras induced by the master equation for a linear chain of coupled harmonic oscillators. The special case of two coupled oscillators presents a family of models that has been the subject of intense studies [28,29,30,31,32,33,34,35,36,37] dealing with the dynamics of entanglement in open continuous-variable systems (see, e.g., [38] for a review).

Typically, this model assumes that there is no interaction between the centre-of-mass and relative-distance modes in the course of relaxation, so that the relaxation part of the Lindbladian is a sum of two commuting relaxation superoperators. By contrast to mechanical systems, in photonic systems with quantised polarisation modes interacting with an optically anisotropic environment, both the dynamical and environment-induced intermode couplings can be important as they manifest themselves in the birefringence and dichroism of absorption [39,40,41]. In this paper, we shall relax this assumption to explore the effects induced by intermode couplings arising from the interaction between the system and the environment. The effects related to the dynamics of entanglement will be our primary concern.

Entanglement is known to be a resource of vital importance for rapidly developing quantum technologies, such as quantum communications and quantum computations [42,43], and its dynamics have been extensively studied during the last two decades (a recent review can be found in [44]). For the cases of discrete and continuous variables, which are typically represented by two interacting oscillators and qubits in the pioneering papers [45,46], it was found that the decay time of entanglement (the time of disentanglement) may be shorter than the time of decoherence and, under certain conditions, revivals of entanglement may occur. Finite-time disentanglement, commonly known as the “sudden death of entanglement”, is another important effect [47,48,49,50,51,52].

The structure of this paper is as follows: In Section 2, after introducing the GKSL master equation for the multi-mode bosonic system interacting with the thermal bath, in Section 2.1, we derive dynamical equations for the mean values of operators that preserve their normal ordering. In Section 2.2, the method of characteristics is utilised to solve the dynamical equation for the normally ordered characteristic function, $\chi_{N}$. In Section 3, the general theoretical results are applied to the important special case of a two-mode bosonic system in the thermal bath. This system can be regarded as a model describing the propagation of quantised polarisation modes in an optical fibre [41]. In Section 3.1, we derive the analytical results needed to evaluate time dependence of the averages that entre the elements of the covariance matrix. The logarithmic negativity of the Gaussian states introduced in Section 3.2 is numerically studied in Section 3.3. Finally, in Section 4, we discuss the results and make some concluding remarks.

## 2. Lindblad Dynamics

### 2.1. Master Equation

The starting point of our analysis is the Markovian thermal bath version of the Lindblad equation for the density matrix of an *N*-mode bosonic system:(1)$$\begin{array}{cc} \; & \frac{\partial \hat{\rho }}{\partial t}=L\hat{\rho }=-i\sum_{n,m=1}^{N}\Omega_{nm}C_{{\hat{a}}_{n}^{†}{\hat{a}}_{m}}\hat{\rho }\\ \; & +\sum_{n,m=1}^{N}K_{nm}\left(D_{{\hat{a}}_{m}{\hat{a}}_{n}^{†}}\hat{\rho }+e^{-z_{T}}D_{{\hat{a}}_{n}^{†}{\hat{a}}_{m}}\hat{\rho }\right) \end{array}$$
written in terms of two superoperators given by
(2)$$\begin{array}{cc} \; & C_{\hat{A}\hat{B}}:\hat{\rho }\mapsto C_{\hat{A}\hat{B}}\hat{\rho }=\left[\hat{A}\hat{B},\hat{\rho }\right],\\ \; & D_{\hat{A}\hat{B}}:\hat{\rho }\mapsto D_{\hat{A}\hat{B}}\hat{\rho }=2\hat{A}\hat{\rho }\hat{B}-\hat{B}\hat{A}\hat{\rho }-\hat{\rho }\hat{B}\hat{A} \end{array}$$
(3)$$\begin{array}{cc} \; & =\left[\hat{A},\hat{\rho }\hat{B}\right]+\left[\hat{A}\hat{\rho },\hat{B}\right], \end{array}$$
where the dagger denotes Hermitian conjugation, $\hat{\rho }$ is the density matrix representing the quantum state; ${\hat{a}}_{n}^{†}$ (${\hat{a}}_{n}$) is the creation (annihilation) operator of the *n*th mode; $\left[\hat{A},\hat{B}\right]=\hat{A}\hat{B}-\hat{B}\hat{A}$ stands for the commutator; $\Omega_{nm}$ ($K_{nm}$) is the element of the frequency (relaxation) matrix, Ω (K); $z_{T}$ is the dimensionless inverse temperature parameter given by
(4)$$\begin{array}{c} z_{T}=\frac{\hbar \Omega_{0}}{k_{B}T}, \end{array}$$
where $\Omega_{0}$ is the bare frequency, *ℏ* is the reduced Planck constant, $k_{B}$ is the Boltzmann constant and *T* is the temperature of the environment. The frequency and relaxation matrices are both Hermitian: $\Omega =\Omega^{†}$ and $K=K^{†}$. The relaxation matrix K with elements giving the rates of thermalization is also positive definite: K>0.

Note that the system (Equation 1) is conveniently rewritten in the following alternative form:(5)$$\begin{array}{cc} \; & \frac{\partial \hat{\rho }}{\partial t}=-i\left[\hat{\Omega },\hat{\rho }\right]-\left(1+e^{-z_{T}}\right)\left[\hat{K},\hat{\rho }\right]\\ \; & +2\sum_{n,m}K_{nm}\left(\left[{\hat{a}}_{m},\hat{\rho }{\hat{a}}_{n}^{†}\right]+e^{-z_{T}}\left[{\hat{a}}_{n}^{†},\hat{\rho }{\hat{a}}_{m}\right]\right), \end{array}$$
where the operators
(6)$$\begin{array}{cc} \; & \hat{\Omega }=\sum_{n,m}\Omega_{nm}{\hat{a}}_{n}^{†}{\hat{a}}_{m},\;\hat{K}=\sum_{n,m}K_{nm}{\hat{a}}_{n}^{†}{\hat{a}}_{m} \end{array}$$
are determined by the frequency and relaxation matrices, Ω and Γ, through the Jordan mapping that maps a Hermitian matrix *J* to the quadratic boson operator $\hat{J}$ as follows:(7)$$\begin{array}{c} J\mapsto \hat{J}=\sum_{n,m}J_{nm}{\hat{a}}_{n}^{†}{\hat{a}}_{m}. \end{array}$$

Our next step is to deduce the dynamic equation for the mean value of an operator $\hat{S}$: $\langle \hat{S}\rangle =\mathrm{Tr}\left(\hat{S}\hat{\rho }\right)$. From Equation (Equation 1) combined with the algebraic identities
(8)$$\begin{array}{cc} \; & \mathrm{Tr}\left(\hat{S}C_{\hat{A}\hat{B}}\hat{\rho }\right)=\langle \left[\hat{S},\hat{A}\right]\hat{B}\rangle +\langle \hat{A}\left[\hat{S},\hat{B}\right]\rangle , \end{array}$$
(9)$$\begin{array}{cc} \; & \mathrm{Tr}\left(\hat{S}D_{\hat{A}\hat{B}}\hat{\rho }\right)=\langle \hat{B}\left[\hat{S},\hat{A}\right]\rangle -\langle \left[\hat{S},\hat{B}\right]\hat{A}\rangle \end{array}$$
we have
(10)$$\begin{array}{cc} \; & \frac{\partial \langle \hat{S}\rangle }{\partial t}=\sum_{n,m}\left(i\Omega_{nm}-\Gamma_{nm}\right)\langle {\hat{a}}_{n}^{†}\left[{\hat{a}}_{m},\hat{S}\right]\rangle \\ \; & -\sum_{n,m}\left(i\Omega_{nm}+\Gamma_{nm}\right)\langle \left[\hat{S},{\hat{a}}_{n}^{†}\right]{\hat{a}}_{m}\rangle \\ \; & +2\sum_{n,m}\Gamma_{nm}n_{T}\langle \left[{\hat{a}}_{m},\left[\hat{S},{\hat{a}}_{n}^{†}\right]\right]\rangle , \end{array}$$
where $\Gamma_{nm}=\left(1-e^{-z_{T}}\right)K_{nm}$ and $n_{T}={\left(e^{z_{T}}-1\right)}^{-1}$ is the mean number of thermal photons.

An important point is that, for a normally ordered operator $\hat{S}$ with $\hat{S}=:\hat{S}:$, the algebraic operations that entre the averages on the right hand side of Equation (Equation 10) preserve normal ordering.

### 2.2. Characteristic Function

In this section, we concentrate on the normally ordered characteristic function given by
(11)$$\begin{array}{c} \chi_{N}\left(\alpha \right)=\langle e^{\left(\alpha ,{\hat{a}}^{†}\right)}e^{-(\alpha^{*},\hat{a})}\rangle , \end{array}$$
where
(12)$$\begin{array}{c} \left(\alpha^{*},\hat{a}\right)\equiv \sum_{i=1}^{N}\alpha_{i}^{*}{\hat{a}}_{i},\;\left(\alpha ,{\hat{a}}^{†}\right)\equiv \sum_{i=1}^{N}\alpha_{i}{\hat{a}}_{i}^{†}, \end{array}$$

$f\left(\alpha \right)\equiv f\left(\alpha_{1},\cdots ,\alpha_{N},\alpha_{1}^{*},\cdots ,\alpha_{N}^{*}\right)$ and an asterisk will indicate complex conjugation.

By using Equation (Equation 10), it is rather straightforward to deduce the following equation for $\chi_{N}$: (13)$$\begin{array}{cc} \; & \frac{\partial \chi_{N}\left(\alpha ,t\right)}{\partial t}=\hat{L}\chi_{N}\left(\alpha ,t\right)\\ \; & \hat{L}=i\sum_{n,m=1}^{N}\Omega_{nm}{\hat{D}}_{mn}^{\left(-\right)} \end{array}$$(14)$$\begin{array}{cc} \; & -\sum_{n,m=1}^{N}\Gamma_{nm}\left({\hat{D}}_{mn}^{\left(+\right)}+2n_{T}\alpha_{m}\alpha_{n}^{*}\right), \end{array}$$
where
(15)$$\begin{array}{c} {\hat{D}}_{mn}^{\left(\pm \right)}=\alpha_{m}\frac{\partial }{\partial \alpha_{n}}\pm \alpha_{n}^{*}\frac{\partial }{\partial \alpha_{m}^{*}}. \end{array}$$

The temporal evolution of the characteristic function $\chi_{N}$ is governed by the dynamical Equation (Equation 13) supplemented with the initial condition
$$\begin{array}{cc} \; & \chi_{N}\left(\alpha ,0\right)\equiv \chi_{\mathrm{ini}}\left(\alpha \right) \end{array}$$
(16)$$\begin{array}{cc} \; & =\int d\mu \left(\beta \right)e^{\left[\left(\alpha^{*},\beta \right)-\left(\alpha ,\beta^{*}\right)\right]}P\left(\beta ,0\right), \end{array}$$
(17)$$\begin{array}{cc} \; & d\mu \left(\beta \right)=\prod_{n=1}^{N}d\mu \left(\beta_{i}\right),\;d\mu \left(\beta_{i}\right)=\frac{d^{2}\beta_{i}}{\pi }, \end{array}$$
where
(18)$$\begin{array}{c} \left(\alpha^{*},\beta \right)\equiv \sum_{i=1}^{N}\alpha_{i}^{*}\beta_{i} \end{array}$$
and $\chi_{N}$ is expressed in terms of the Glauber–Sudarshan *P* function (quasidistribution), P(β,t), related to the *P*-representation of the density matrix as follows:(19)$$\begin{array}{cc} \; & \hat{\rho }\left(t\right)=\int d\mu \left(\beta \right)P\left(\beta ,t\right)|\beta \rangle \langle \beta |. \end{array}$$

We can now employ the method of characteristics [53] to solve the above initial value problem. According to this method, we begin with the system of characteristic equations
(20)$$\begin{array}{c} \frac{\partial \alpha_{n}}{\partial t}=-\sum_{m=1}^{N}Q_{nm}\;\alpha_{m},\;Q=i\Omega -\Gamma \end{array}$$
and its solution written in the matrix form as follows:(21)$$\begin{array}{c} \alpha =e^{-Qt}\alpha_{0}\equiv A\left(-t\right)\alpha_{0},\;A\left(t\right)=e^{(i\Omega -\Gamma )t}\equiv e^{Qt}, \end{array}$$
where $\alpha_{0}=\alpha \left(0\right)$. It is not difficult to obtain the solution along the characteristic curves (Equation 21) given by
(22)$$\begin{array}{cc} \; & \chi_{N}\left(\alpha_{0},t\right)=\exp\left[-2n_{T}\int_{0}^{t}\left(\alpha_{0}^{*},A^{†}\left(-\tau \right)\Gamma A\left(-\tau \right)\alpha_{0}\right)d\tau \right]\\ \; & \times \chi_{\mathrm{ini}}\left(\alpha_{0}\right). \end{array}$$

We can now express $\alpha_{0}$ in terms of α with the help of Equation (Equation 21): $\alpha_{0}=A\left(t\right)\alpha$ and use the identity
(23)$$\begin{array}{c} \frac{d}{d\tau }A^{†}\left(-\tau \right)A\left(-\tau \right)=2A^{†}\left(-\tau \right)\Gamma A\left(-\tau \right), \end{array}$$
to transform Formula (Equation 22) into the final expression for the characteristic function: (24)$$\begin{array}{cc} \; & \chi_{N}\left(\alpha ,t\right)=e^{-(\alpha^{*},B\left(t\right)\alpha )}\chi_{\mathrm{ini}}\left(A\left(t\right)\alpha \right), \end{array}$$(25)$$\begin{array}{cc} \; & B\left(t\right)=n_{T}\left(I_{N}-A^{†}\left(t\right)A\left(t\right)\right), \end{array}$$
where $I_{N}$ is the identity N×N matrix. This formula is our central analytical result that will be used in the subsequent sections.

## 3. Two-Mode System

In this section, we focus our attention on the task of computing the time dependence of the averages that can be regarded as one-point correlation functions. One of the approaches to this important problem is to derive and solve dynamical equations for the averaged operators. For example, Equation (Equation 10) and the algebraic identities for the bosonic bilinear forms (Equation 7)
(26)$$\begin{array}{cc} \; & \sum_{n,m}\Gamma_{nm}\langle {\hat{a}}_{n}^{†}\left[{\hat{a}}_{m},\hat{J}\right]\rangle =\sum_{n,m}{\left(J\Gamma \right)}_{nm}{\hat{a}}_{n}^{†}{\hat{a}}_{m}\equiv \hat{J\Gamma },\\ \; & \sum_{n,m}\Gamma_{nm}\langle \left[\hat{J},{\hat{a}}_{n}^{†}\right]{\hat{a}}_{m}\rangle =\sum_{n,m}{\left(\Gamma J\right)}_{nm}{\hat{a}}_{n}^{†}{\hat{a}}_{m}\equiv \hat{\Gamma J} \end{array}$$
can be utilised to deduce the equation for the mean value $\langle \hat{J}\rangle$ given by
(27)$$\begin{array}{cc} \; & \frac{\partial \langle \hat{J}\rangle }{\partial t}=-i\langle \hat{\left[J,\Omega \right]}\rangle -\langle \hat{\left\{J,\Gamma \right\}}\rangle +2n_{T}\mathrm{Tr}\left(J\Gamma \right), \end{array}$$
where {A,B}=AB+BA denotes the anticommutator.

For illustrative purposes, in what follows, we consider a photonic system with two orthogonally polarised quantised modes [40,41,54], thus restricting our analysis to the special case of the two-mode bosonic system with N=2. Then, the above general result can be applied to the so-called Stokes operators that can be expressed in terms of the Pauli matrices
(28)$$\begin{array}{cc} \; & \sigma_{1}=\left(\begin{array}{cc} 0 & 1\\ 1 & 0 \end{array}\right),\;\sigma_{2}=\left(\begin{array}{cc} 0 & -i\\ i & 0 \end{array}\right),\;\sigma_{3}=\left(\begin{array}{cc} 1 & 0\\ 0 & -1 \end{array}\right) \end{array}$$
as follows [55]:(29)$$\begin{array}{c} {\hat{S}}_{i}=\sum_{n,m=1}^{2}\sigma_{nm}^{\left(i\right)}{\hat{a}}_{n}^{†}{\hat{a}}_{m}, \end{array}$$
where 0≤i≤3 and $\sigma_{0}=I_{2}$ is the 2×2 identity matrix. The mean values of these operators are known as the Stokes parameters and describe the state of polarization of the photonic system (dynamical regimes of the Stokes parameters are studied in [54]). These parameters, however, appear to be insufficient for complete characterization of the quantum states that generally requires the knowledge of higher order moments of the Stokes operators [55]. For such moments, the approach based on dynamical equations quickly becomes rather involved and unnecessarily complicated.

### 3.1. Exact Dynamics of Averages and Covariance Matrix

An alternative method is to use a formula for the characteristic Function (Equation 24). The derivatives of this function can be easily evaluated, giving the expressions for the mean values of normally ordered operators. In particular, the second order moments are computed as follows:$$\begin{array}{cc} \; & \langle {\hat{a}}_{i}^{†}{\hat{a}}_{j}\rangle \left(t\right)=-\frac{\partial^{2}\chi_{N}\left(\alpha ,t\right)}{\partial \alpha_{i}\partial \alpha_{j}^{*}}{|}_{\alpha =0} \end{array}$$(30)$$\begin{array}{cc} \; & =B_{ji}\left(t\right)+\sum_{n,m}A_{ni}\left(t\right)A_{mj}^{*}\left(t\right)\langle {\hat{a}}_{n}^{†}{\hat{a}}_{m}\rangle \left(0\right)\\ \; & \langle {\hat{a}}_{i}^{†}{\hat{a}}_{j}^{†}\rangle \left(t\right)=\frac{\partial^{2}\chi_{N}\left(\alpha ,t\right)}{\partial \alpha_{i}\partial \alpha_{j}}{|}_{\alpha =0} \end{array}$$(31)$$\begin{array}{cc} \; & =\sum_{n,m}A_{ni}\left(t\right)A_{mj}\left(t\right)\langle {\hat{a}}_{n}^{†}{\hat{a}}_{m}^{†}\rangle \left(0\right). \end{array}$$

These moments determine the elements of the covariance matrix of our two-mode photonic system [56] and the relations (Equation 30) and (31) yield the starting point of our analysis in the subsequent section.

For the two-mode system, the frequency and relaxation matrices can be written as a linear combination of the Pauli matrices
(32)$$\begin{array}{cc} \; & i\Omega -\Gamma =\sum_{k=0}^{3}\left(i\omega_{k}-\gamma_{k}\right)\sigma_{k}=\left(i\omega_{0}-\gamma_{0}\right)\sigma_{0}\\ \; & +(i\omega -\gamma ,\sigma ), \end{array}$$
where $\sigma \equiv (\sigma_{1},\sigma_{2},\sigma_{3})$; $\omega \equiv (\omega_{1},\omega_{2},\omega_{3})$ and $\gamma \equiv (\gamma_{1},\gamma_{2},\gamma_{3})$ are the frequency and the relaxation rate vector, respectively; and it is rather straightforward to obtain the matrix exponential for A(t) in the following explicit form:(33)$$\begin{array}{cc} \; & A\left(t\right)=e^{(i\omega_{0}-\gamma_{0})t}\{\cosh\left(\left(\gamma -i\omega \right)t\right)\sigma_{0}\\ \; & +\frac{\sinh((\gamma -i\omega )t)}{\gamma -i\omega }\left(i\omega -\gamma ,\sigma \right)\}, \end{array}$$
where
(34)$$\begin{array}{c} \gamma -i\omega =\sqrt{\left(\gamma -i\omega ,\gamma -i\omega \right)}=\sqrt{\sum_{k=1}^{3}{\left(\gamma_{k}-i\omega_{k}\right)}^{2}}. \end{array}$$

We can now use Formulas (Equation 30)–(Equation 34) to evaluate the covariance matrix, Σ, of our two-mode bosonic system. This matrix can be defined in terms of the quadrature operators (quadratures), ${\hat{x}}_{i}$ and ${\hat{p}}_{i}$, expressed in terms of the annihilation and creation operators, ${\hat{a}}_{i}$ and ${\hat{a}}_{i}^{†}$, as follows (see, e.g., references [38,56]):(35)$$\begin{array}{c} {\hat{r}}_{i}\equiv \left(\begin{array}{c} {\hat{x}}_{i}\\ {\hat{p}}_{i} \end{array}\right)=C\left(\begin{array}{c} {\hat{a}}_{i}\\ {\hat{a}}_{i}^{†} \end{array}\right),\;C=\frac{1}{\sqrt{2}}\left(\begin{array}{cc} 1 & 1\\ -i & i \end{array}\right). \end{array}$$

In our case, the block structure of the covariance matrix is given by
(36)$$\begin{array}{cc} \; & \Sigma =\left(\begin{array}{cc} \Sigma_{11} & \Sigma_{12}\\ \Sigma_{21} & \Sigma_{22} \end{array}\right),\;\Sigma_{ij}=\Sigma_{ij}^{\left(0\right)}-2\langle {\hat{r}}_{i}\rangle ⊗\langle {\hat{r}}_{j}\rangle , \end{array}$$
(37)$$\begin{array}{cc} \; & \Sigma_{ij}^{\left(0\right)}=\left(\begin{array}{cc} \langle \left\{{\hat{x}}_{i},{\hat{x}}_{j}\right\}\rangle & \langle \left\{{\hat{x}}_{i},{\hat{p}}_{j}\right\}\rangle \\ \langle \left\{{\hat{p}}_{i},{\hat{x}}_{j}\right\}\rangle & \langle \left\{{\hat{p}}_{i},{\hat{p}}_{j}\right\}\rangle \end{array}\right), \end{array}$$
where the diagonal and off-diagonal block matrices
(38)Σii(0)=2〈a^i†a^i〉+1+2Re〈a^i2〉2Im〈a^i2〉2Im〈a^i2〉2〈a^i†a^i〉+1−2Re〈a^i2〉,Σ12(0)=2C〈a^1a^2†〉〈a^1a^2〉〈a^1†a^2†〉〈a^1†a^2〉C†=x1+x2y1+y2−y1+y2x1−x2,Σ21=Σ12T,〈a^1†a^2〉=x1+iy12,〈a^1a^2〉=x2+iy22
are expressed in terms of the averages: $\langle {\hat{a}}_{i}^{†}{\hat{a}}_{j}\rangle$ and $2\langle {\hat{a}}_{i}{\hat{a}}_{j}\rangle$.

### 3.2. Symplectic Eigenvalues and Logarithmic Negativity

In what follows, we shall restrict our analysis to the important special case of the Gaussian states. These states are characterised by the symplectic eigenvalues of the covariance matrix (Equation 36) that can be computed using the relations [38,56]:(39)$$\begin{array}{cc} \; & 2\mu_{\pm }^{2}=\Delta \pm \sqrt{\Delta^{2}-4\det\Sigma }, \end{array}$$
where $\Delta =\det\Sigma_{11}+\det\Sigma_{22}+2\det\Sigma_{12}$ is known as the “seralian” invariant.

The separability and entanglement criteria for bipartite continuous variable systems formulated in terms of the covariance matrix [29,57,58] and more general criteria [59,60] involving higher-order correlators provide a number of entanglement witnesses. For example, such witnesses are used for the analysis of the experimental data presented in [61] and the coherent state quantum key distribution suggested in [62,63].

For two-mode Gaussian states, different measures of entanglement have been proposed. These include the entanglement of formation, the Bures distance and the Gaussian measures of entanglement [64,65,66]. In this work, we deploy the logarithmic negativity [56] as a useful quantifier of bipartite entanglement in Gaussian states given by
(40)$$\begin{array}{c} E_{N}\left(\hat{\rho }\right)=\log_{2}\left|\right|{\hat{\rho }}^{PT}{\left|\right|}_{1}=\max\left\{0,-\log_{2}\lambda_{-}\right\}, \end{array}$$
where ${\left||.|\right|}_{1}$ stands for the trace norm and ${\hat{\rho }}^{PT}$ is the partial transpose of $\hat{\rho }$. The right-hand side of Equation (Equation 40) gives the expression for $E_{N}\left(\hat{\rho }\right)$ in terms of the lowest symplectic eigenvalue $\lambda_{-}$ of the partially transposed density matrix, ${\hat{\rho }}^{PT}$, with the symplectic eigenvalues given by
(41)$$\begin{array}{cc} \; & 2\lambda_{\pm }^{2}=\Delta_{-}\pm \sqrt{\Delta_{-}^{2}-4\det\Sigma }, \end{array}$$
where $\Delta_{-}=\det\Sigma_{11}+\det\Sigma_{22}-2\det\Sigma_{12}$. The logarithmic negativity being an entanglement monotone (a quantity which cannot be increased using local operations and classical communication) is known to bound the distillable entanglement contained in $\hat{\rho }$ [67]. Note that, recently, the link between the logarithmic negativity and the quadrature coherence scale introduced as a nonclassicality measure was studied in [68].

### 3.3. Numerical Results

We assume that the system is initially prepared in the two-mode squeezed state
(42)$$\begin{array}{c} {|\eta \rangle }_{\mathrm{sq}}=\exp\left[\eta {\hat{K}}_{+}-\eta^{*}{\hat{K}}_{-}\right]|0\rangle ,\;\eta =re^{i\theta }, \end{array}$$
where *r* is the squeezing parameter, ${\hat{K}}_{+}={\hat{a}}_{1}^{†}{\hat{a}}_{2}^{†}$ (${\hat{K}}_{-}={\hat{a}}_{1}{\hat{a}}_{2}$) is the raising (lowering) generator of the su(1,1) Lie algebra, with the covariance matrix given by
(43)$$\begin{array}{cc} \; & \Sigma_{11}\left(0\right)=\Sigma_{22}\left(0\right)=\cosh\left(2r\right)\sigma_{0},\\ \; & \Sigma_{12}\left(0\right)=\sinh\left(2r\right)\left[\cos\left(\theta \right)\sigma_{3}+\sin\left(\theta \right)\sigma_{1}\right]. \end{array}$$

According to Equation (Equation 33), the frequency and the relaxation rate vectors, ω and γ, describe the dynamical and environment mediated intermode couplings, respectively. These vectors determine the regime of the dissipative dynamics and will be parametrised using the angular representation:(44)$$\begin{array}{cc} \; & \gamma =\Gamma (\sin\theta_{\Gamma }\cos\phi_{\Gamma },\sin\theta_{\Gamma }\sin\phi_{\Gamma },\cos\theta_{\Gamma }),\\ \; & \omega =\Omega (\sin\theta_{\Omega }\cos\phi_{\Omega },\sin\theta_{\Omega }\sin\phi_{\Omega },\cos\theta_{\Omega }). \end{array}$$

In order to investigate the effects of the intermode couplings in the dynamics of entanglement, we have computed the time dependencies of the logarithmic negativity for differently oriented vectors γ and ω. The list of parameters used in our calculations is as follows: $n_{T}=0.1$, r=2.0, $\Gamma /\gamma_{0}=0.9$ and $\theta =\phi_{\Gamma }=\phi_{\Omega }=0$${}^{\circ }$. At these parameters, we have the initial values of the symplectic eigenvalues $\mu_{\pm }\left(0\right)=1$, $\mu_{1,2}\left(0\right)=\cosh\left(2r\right)$ and $\lambda_{-}\left(0\right)=e^{-2r}$ giving the corresponding value for the logarithmic negativity, $E_{N}\left(\hat{\rho }\right){|}_{t=0}\approx 5.77$.

For the dynamics governed by the thermal bath Lindblad Equation (Equation 1), in the long time limit with t→∞, the density matrix approaches the equilibrium state ${\hat{\rho }}_{\mathrm{eq}}=\hat{\rho }\left(\infty \right)$ with the covariance matrix given by
(45)$$\begin{array}{c} \Sigma \left(t\right)\overset{t\to \infty }{\to }\Sigma_{\mathrm{eq}}=\left(1+2n_{T}\right)I_{4}. \end{array}$$

The equilibrium density matrix, ${\hat{\rho }}_{\mathrm{eq}}\propto \exp\left[-z_{T}\left({\hat{a}}_{1}^{†}{\hat{a}}_{1}+{\hat{a}}_{2}^{†}{\hat{a}}_{2}\right)\right]$, is the disentangled mixed Gaussian state with $E_{N}\left({\hat{\rho }}_{\mathrm{eq}}\right)=I\left(1;2\right){|}_{t\to \infty }=0$ and $S\left[{\hat{\rho }}_{\mathrm{eq}}\right]\equiv S_{\mathrm{eq}}\approx 0.56$.

In Figure 1, we present the results on the temporal evolution of the logarithmic negativity computed at Ω=0 for different values of the angle $\theta_{\Gamma }$. The curves clearly indicate the regime of finite-time disentanglement known as the “sudden death of entanglement”. In this regime, $E_{N}$ vanishes at $t\geq t_{d}$, where $t_{d}$ is the time of disentanglement.

Figure 2 shows the disentanglement time calculated as a function of $\theta_{\Gamma }$. It can be seen that $t_{d}$ is sensitive to orientation of the relaxation rate vector describing the intermode coupling induced by the interaction between the bosonic system and the environment.

The curves plotted in Figure 3 are computed at different values of the angle $\theta_{\Omega }$ that specify the orientation of the frequency vector with $\Omega /\gamma_{0}=\pi$ and illustrate the effect of the dynamical intermode coupling on entanglement dynamics. It is seen that oscillatory behaviour of the logarithmic negativity at $\theta_{\Omega }=90$${}^{\circ }$ translates into sudden revivals of entanglement as the angle $\theta_{\Omega }$ decreases.

## 4. Conclusions

In this paper, we have studied the dynamics of a multi-mode bosonic system governed by the thermal bath Lindblad master equation. Our general approach is based on an exact solution for the characteristic function obtained using the method of characteristics.

We have applied this approach to the special case of the two-mode bosonic system. In this case, the dynamics are determined by the intermode couplings that entre the dynamical and relaxation parts of the Lindblad superoperator L (see Equation (Equation 1)) and can be described in terms of the frequency and the relaxation rate vectors (see Equation (Equation 32)). We have focussed our attention on the effects of the intermode couplings in the dynamics of entanglement and have presented a number of the numerical results on time dependence of the logarithmic negativity.

It is found that, for a vanishing frequency vector with Ω=0, the logarithmic negativity of the system initially prepared in the two-mode squeezed state monotonically decays, reaching zero at the disentanglement point in time, $t=t_{d}$. In this regime of finite-time disentanglement, the disentanglement time appears to be dependent on the orientation of the relaxation rate vector. We have also shown that the presence of the dynamical intermode coupling with Ω>0 complicates the dynamics of the logarithmic negativity, and its nonmonotonic behaviour results in revivals of entanglement.

Our theoretical considerations were motivated by the model describing mixed polarisation quantum states propagating in an optically anisotropic lossy environment [40,41]. Interestingly, our results are formulated in terms of the covariance matrix that can, in principle, be extracted from experimental data measured using either optical homodyne or heterodyne detection techniques [69]. Qualitatively, our results on the regime of finite-time disentanglement and revivals of entanglement are in agreement with the predictions for oscillators interacting with the thermal bath previously published in [28,29,32,33]. It turns out that the values of the disentaglement time reported in [29] and obtained in the limit of negligible relaxation anisotropy where Γ=0 are close to our estimate $\gamma_{0}t_{d}\approx 0.6$ evaluated at $\theta_{\Gamma }=0$.

Our findings provide further insights about the protection techniques of entangled states from the detrimental effects of surrounding environments by suitably manipulating the intermode interaction. Generally, revivals of quantum correlations in composite quantum systems are a useful dynamical feature against these effects [70,71,72,73]. Different experimental methods and theoretical approaches to protect quantum resources have been put forward in [74,75,76,77,78,79,80,81].

We conclude with the remark that our analytical approach can be readily extended to study a number of problems, such as Gaussian Einstein–Podolsky–Rosen steering for two-mode squeezed states transmitted in lossy quantum communication channels [82] and the problems related to controlled quantum dynamics in a realistic setup involving open environments. Quantum navigation is an important class of controlled quantum dynamics whereby the objective is to transport one quantum state into another, or to generate quantum gates, in the shortest possible time under the influence of an uncontrollable external field. Problems of this kind can be thought of as representing the quantum counterpart of the classical Zermelo navigation problem of finding the time-optimal control that takes a ship from one location to another, under the influence of external wind or currents [83,84,85]. In a forthcoming publication, we will apply our results to the Zermelo navigation problem for open multi-mode bosonic systems.

## Figures and Tables

**Figure 1 entropy-23-01409-f001:**
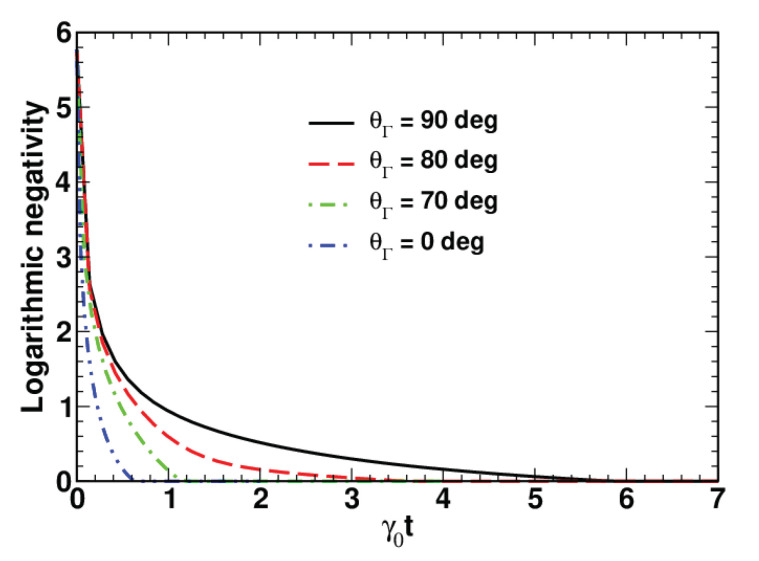
Time dependence of the logarithmic negativity computed at different angles $\theta_{\Gamma }$ with $\gamma =\Gamma (\sin\theta_{\Gamma },0,\cos\theta_{\Gamma })$ for $\Gamma /\gamma_{0}=0.9$, $\Omega /\gamma_{0}=0$, $n_{T}=0.1$, r=2 and θ=0${}^{\circ }$.

**Figure 2 entropy-23-01409-f002:**
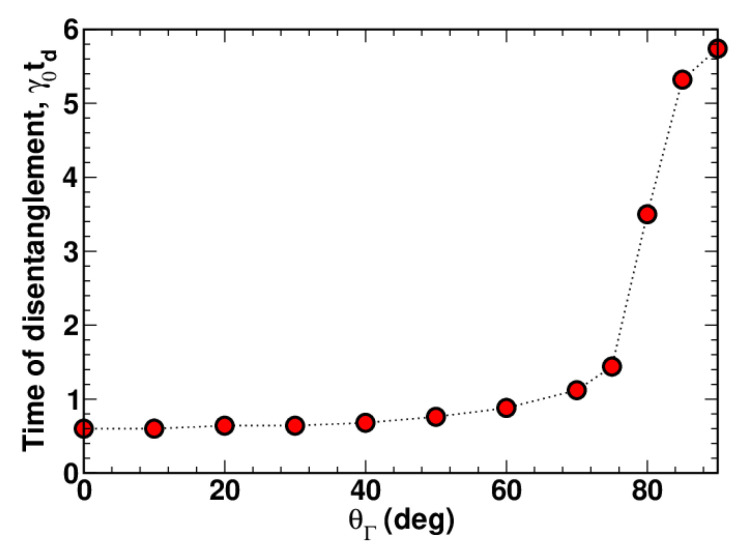
Time of disentanglement as a function of the angle $\theta_{\Gamma }$ computed at $\Gamma /\gamma_{0}=0.9$, and $\Omega /\gamma_{0}=0$ (other parameters are listed in the caption of Figure 1).

**Figure 3 entropy-23-01409-f003:**
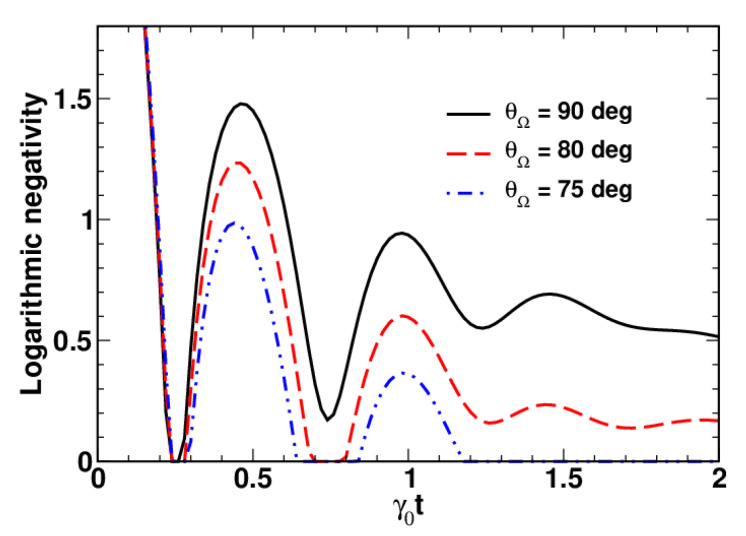
Time dependence of the logarithmic negativity computed at different angles $\theta_{\Omega }$ with $\omega =\Omega (\sin\theta_{\Omega },0,\cos\theta_{\Omega })$ for $\Gamma /\gamma_{0}=0.9$, $\theta_{\Gamma }=90$${}^{\circ }$, $\Omega /\gamma_{0}=\pi$, $n_{T}=0.1$, r=2 and θ=0${}^{\circ }$.

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
