# Peer review of "Lindblad Dynamics and Disentanglement in Multi-Mode Bosonic Systems"

_entropy, 2021, doi:10.3390/e23111409_

Round 1
Reviewer 1 Report
The manuscript entitled "Lindblad dynamics and disentanglement in multi-mode bosonic systems" by Alexei D. Kiselev, Ali Ranim, and Andrei V. Rybin is an original research paper devoted to a study of quantum nonunitary dynamics of quantum states in a multi-mode bosonic system using the thermal bath Lindblad master equation.
First, the authors prepared an extensive review in the field of open quantum systems. Discussing the well-known Gorini-Kossakowski-Sudarshan-Lindblad (Lindblad) equation they expand the model to the case of the interacting modes during relaxation. This treatment possesses them to describe the effects related to the dynamics of entanglement.
The authors described the important case of a two-mode bosonic system in the thermal bath. This system can be regarded as a model describing the propagation of quantized polarization modes in an optical fiber. Further, they derived analytical results for the logarithmic negativity of Gaussian states.
The main results of the paper are the following. The authors discussed how the revivals of entanglement can be induced by the dynamic coupling between the different modes. For the system initially prepared in a two-mode squeezed state, they found the logarithmic negativity as defined by the magnitude and orientation of the frequency and the relaxation rate vectors. They show that, in the regime of finite-time-disentanglement, reorientation of the relaxation rate vector may significantly increase the time of disentanglement.
The paper is of broad interest to the research community in the field of quantum optics and quantum information theory. The obtained results can be useful in solving the important problem of navigation of open quantum systems.
I think that the paper "Lindblad dynamics and disentanglement in multi-mode bosonic systems" by Alexei D. Kiselev, Ali Ranim, and Andrei V. Rybin can be accepted for publication in the Entropy in its present form.
Author Response
-
Reviewer 2 Report
The paper entitled „Lindblad dynamics and disentanglement in multi-mode bosonic systems” is devoted to the application of the master equation describing the interaction of a bosonic system with a Markovian multimode reservoir for the description of entanglement decay.
The problem of entanglement evolution and its decay due to the interactions with the external environment is not new and has been a subject of many papers. The Authors use a well-known Lindblad master equation for reduced density matrix and their idea is to incorporate to the description also the effects that are caused by couplings between the modes in the environment and system-environment.
Using the method of characteristics they obtained analytical expressions leading to the solutions of characteristic functions. The lindbladian reservoir is described by frequency and relaxation rate vectors and in such an approach different variants of couplings between the environmental and system modes can be introduced. The method is tested on a two-mode bosonic system and the analysis of logarithmic negativities, particularly the influence of parameters characterizing frequency and relaxation rate vectors on disentanglement times are studied. It is shown whether one can expect to obtain so-called entanglement „sudden death”.
In my opinion, the manuscript is interesting, presents new findings in studies of open system dynamics, and can be published in Entropy after some amendments to the paper are made.
-) As the problems of disentanglement caused by the environment and sudden death of entanglement in bosonic systems are widely presented in the literature I would expect that the Authors refer somehow to the results that are already known (especially to those they refer to in the introduction). Are their results concerning disentanglement times consistent with them?
-) Could the Authors give some suggestions for possible experimental realizations in which their results can be verified?
-) In the introduction the Authors should emphasize more why the generalization they propose is relevant.
Round 2
Reviewer 2 Report
The authors incorporated all the suggestions from my report. I recommend publication in present form.
